# Public involvement and engagement in big data research: protocol for a scoping review and a systematic review of delivery and effectiveness of strategies for involvement and engagement

Piotr Teodorowski [iD] ,[1] Elisa Jones,[1] Naheed Tahir,[2] Saiqa Ahmed,[2] Lucy Frith [iD] [3]

¹Institute of Population Health, University of Liverpool, Liverpool, UK
²ARC NWC Public Advisor, Liverpool, UK
³Departments of Law and Philosophy, University of Liverpool, Liverpool, UK

**Correspondence to**
Piotr Teodorowski;
p.teodorowski@liverpool.ac.uk

## ABSTRACT

**Introduction** Big data research has grown considerably over the last two decades. This presents new ethical challenges around consent, data storage and anonymisation. Big data research projects require public support to succeed and it has been argued that one way to achieve this is through public involvement and engagement. To better understand the role public involvement and engagement can play in big data research, we will review the current literature. This protocol describes the planned review methods.

**Methods and analysis** Our review will be conducted in two stages. In the first stage, we will conduct a scoping review using Arksey and O'Malley methodology to comprehensively map current evidence on public involvement and engagement in big data research. Databases (CINAHL, Health Research Premium Collection, PubMed, Scopus, Web of Science) and grey literature will be searched for eligible papers. We provide a narrative description of the results based on a thematic analysis. In the second stage, out of papers found in the scoping review which discuss involvement and engagement strategies, we will conduct a systematic review following Preferred Reporting Items for Systematic Reviews and Meta-Analyses guidelines, exploring the delivery and effectiveness of these strategies. We will conduct a qualitative synthesis. Relevant results from the quantitative studies will be extracted and placed under qualitative themes. Individual studies will be appraised through Mixed Methods Appraisal Tool (MMAT), we will then assess the overall confidence in each finding through Confidence in the Evidence from Reviews of Qualitative research (GRADE-CERQual). Results will be reported in a thematic and narrative way.

**Ethics and dissemination** This protocol sets out how the review will be conducted to ensure rigour and transparency. Public advisors were involved in its development. Ethics approval is not required. Review findings will be presented at conferences and published in peer-reviewed journals.

## Strengths and limitations of this study

► This is the first review exploring public involvement and engagement in big data research.
► The search is limited to studies published in English.
► Lack of clarity and consistency with the use of the terms public involvement, engagement and big data could impact our search results. However, we will undertake additional searching techniques to mitigate this limitation.

## INTRODUCTION

### What is the problem?

Over the last two decades, the ongoing digitalisation of information has allowed the creation and linkage of large, multi-source health data sets to provide novel healthcare applications. This is often called 'big data', but the concept itself is unclear and heavily debated.[1] However, this growing area of research has the following characteristics: large volume, high velocity, huge variety, veracity and value.[1] Multiple stakeholders use big data for research; clinical management; audit; service evaluation or statistical purposes. The UK has been a global leader in big data research. Large projects include, at national level, OpenSAFELY[2] and regionally located projects such as Children Growing Up in Liverpool[3] (to name a few). The overriding aims of big data research projects are to deliver more efficient healthcare,[4] and to reduce health inequalities.[5]

The use of big data for research presents ethical challenges.[6] Traditionally, a person consents to participate in a research study, whereas when large quantities of data are collected, it is not often apparent how it will be (re)used in the future. Data can be collected for one purpose (eg, audit or to collect groups statistics) and only later shared or linked for

research. Second, even when big data is anonymised, in theory, individuals can be still re-identified.[6] Third, digitalised data needs to be stored—sometimes in various places and hosted by both public institutions and private companies. Despite these ethical issues (consent, anonymisation, data storage and access), the literature shows that the public mostly supports big data usage in research,[7] but is sceptical toward current governance mechanisms[8] and concerned about associated risks such as breach of privacy, generating waste of unused information and usage of data for profit rather than for the public good. Big data is still new, and thus it often outpaces governance structures and regulation. Even if researchers meet the legal requirements, the public might not be supportive of their actions.[9 10] Controversial cases can undermine public trust in big data. For example, the case of DeepMind in the UK illustrated these dangers: the project breached data protection legislation by sharing patients' data (without properly informing them) with the Google-owned company.[11] Low public engagement and lack of transparency in the care.data project in the UK[12] led to its eventual closure. The public might perceive the risk–benefit ratio as unfavourable for them and therefore not want to support or participate in the research. Also, it could foster general distrust in healthcare professionals.

## What is the solution?

The concept of trust is vital in building a positive relationship between researchers and the public.[8] Improving people's knowledge, through public engagement, of how big data research works can improve public support for using health data.[13] For example, the #DataSavesLives initiative raises awareness of the benefits of health data research to gain public trust.[14] Second, researchers should involve the public in developing transparent, accountable policies and governance processes.[15] Public involvement and engagement are crucial mechanisms to develop governance policies and build trust between the public and researchers. Public involvement should be genuine. It should not be carried out with the sole aim of benefiting researchers; be tokenistic or mislead the public.[16] Extensive evidence shows that successful public involvement can lead to service improvement,[17–20] raises awareness of services[20] and brings together patients' and researchers' priorities.[21]

Public involvement in big data research has context-related challenges. In traditional research, a participant and a researcher would have some contact. In contrast, big data research includes large groups of people (who might not necessarily be aware that a particular research team uses their data), thus creating a feeling of remoteness between researchers and the public.[10] Therefore, building trust between the public and researchers is more challenging. Transparent governance policies need to be developed with public involvement to ensure transparency. Lay people can be members of ethics and governance committees overseeing research projects, ensuring public voices are heard. Researchers need big data

specific recommendations on involving and engaging the public. However, the literature on public involvement and engagement in big data research is still limited.

## Why is this review needed?

Systematic and narrative reviews that have explored the public attitudes towards big data have typically focused on trust or attitudes towards using big data for research.[7 22–24] However, how and to what extent public involvement and engagement is used in establishing trust for big data research (eg, organising and maintaining large health data sets and its governance policies) has received less attention. To our knowledge, there is no review covering our objectives published or registered on PROSPERO or Cochrane databases.

To better understand the complexity of public involvement and engagement in big data research, we developed a system logic model (see figure 1) following Rohwer *et al* guidance.[25] Graphic presentations (such as logic models) can be used in reviews to identify relevant elements and the relationships between them. This model is based on team discussion, a preliminary scoping of literature, and public advisors' feedback. We used asterisks (*) to record those sections which were suggested by public advisors. Our model puts special emphasis on four related sections: context, design of public involvement and engagement strategies, targeted population and outcomes. As our review progresses, we will develop the logic model, and present the final version in the report of our review's findings. We hope that the model will assist in interpreting the findings and identifying gaps in the literature.

## Review objectives

The purpose of this review is to synthesise the evidence on public involvement and engagement in big data research. We have two objectives:

► Comprehensively map current evidence on public involvement and engagement in big data research (scoping review).
► Use this to synthesise evidence on the delivery and effectiveness of involvement and engagement strategies (systematic review).

## METHODS AND ANALYSIS
## Design

The review will be conducted in two stages as illustrated in figure 2.[26] These stages will complement each other and assist in flexibly understanding the phenomenon. First, the literature on public involvement and engagement in big data research will be explored by conducting a scoping review. We follow Arksey and O'Malley[27] framework and its further iterations.[28 29] The scoping review will allow us to clarify concepts, illustrate current evidence in the field and gaps in research.[30] In the second stage, out of papers identified in the scoping review, we will extract those discussing involvement and engagement strategies to explore their delivery and effectiveness. The findings from the systematic review will inform researchers on best

**Figure 1** System logic model of public involvement and engagement in big data research. HCP, healthcare professionals; PPI, public and patient involvement.

practice and identify any conflicting views.[30] To further enhance the quality of this review, we follow Preferred Reporting Items for Systematic Reviews and Meta-Analyses (PRISMA) reporting guidelines.[31]

### Stage 1: scoping review
#### Search strategy
We will search the following databases CINAHL, Health Research Premium Collection, PubMed, Scopus, Web of Science and check sources of grey literature related to public involvement such as the Patient-Centred Outcome Research Institute. The first hundred hits (to be inclusive but practical) of Google Scholar search results will be scanned for inclusion. We will also hand-search papers in the journals *Health Expectations*, *BMC Research Involvement and Engagement* and the *International Journal of Population Data Science*. This will be followed by snowball sampling where we will check references in included papers to identify additional studies for inclusion and consult with experts about relevant papers. Big data research is a newly developing field; for instance, MeSH terms 'big data' was added in 2019. Thus, to capture these recent developments, we will restrict searches to a start date of 2010 and will update our searches prior to the final submission of our findings.

We developed the search strategy in partnership with an information specialist and tested this through an iterative process. It consists of both Boolean operators and where possible MeSH (PubMed) or subject heading (CINAHL). Three databases were searched in a test run and yielded a large number of references that were not relevant to our review aims. Therefore, we decided to include the further term 'data governance' as we expect that most of the public involvement and engagement in big data research would be at the stage of developing and maintaining data sets. The summary of the search strategy is presented in table 1.

#### Inclusion and exclusion criteria in the scoping review
Public involvement and engagement can take place at any stage of a big data research project. Thus, we will include papers relating to any public role or contribution to big data research. These roles can include permission to use data, involvement in defining aims or design, and participation in decision-making processes (also the public may become members of a research team).[16]

Previous reviews[17 32 33] have noted that a lack of one generally accepted definition of public involvement makes searching databases challenging. Hence, the definition of public involvement and engagement in the literature lacks consistency.[20] Involvement, engagement, participation are often used interchangeably in the literature but do not necessarily have the same meaning.[34] We follow the INVOLVE definition of public involvement and engagement.[35]

#### Public involvement
'Research being carried out 'with' or 'by' members of the public rather than 'to', 'about' or 'for' them'.

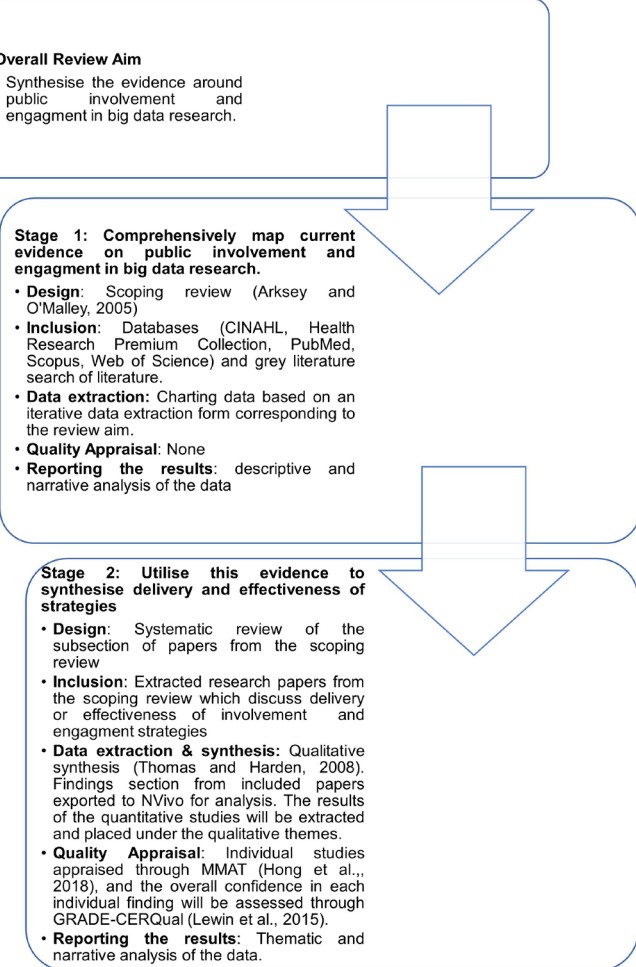

**Figure 2** Systematic map of the review process. GRADE-CERQual, Confidence in the Evidence from Reviews of Qualitative research; MMAT, Mixed Methods Appraisal Tool.

## Consultation

Researchers discussed the project with members of the public. It was more of 'to', 'about' or 'for' rather than 'with' or 'by' them.

## Public engagement

'Information and knowledge about research is provided and disseminated'—this usually takes place after the project is concluded.

INVOLVE's definition of involvement sees an equal relationship between researchers and the public. Thus, involvement should mean codesign and coproduction rather than just consultation. However, we will not exclude papers that do not meet this requirement but note it. Thus, included papers will be assigned one of three named categories: green (when it meets the definition of public involvement), blue (when consultation took place) and amber (where only the engagement occurred).

Multiple definitions of big data exist.[1] To broadly map the current evidence, we use a definition which focuses on big data in the healthcare setting.

| Table 1 | Search strategy |
|---|---|
| Public | "advisory group" OR carer* OR citizen* OR client* OR communit* OR consumer* OR famil* OR lay OR nonpatient* OR participant* OR patient* OR public OR relative* OR representative* OR stakeholder* OR "steering group*" OR survivor* OR user* |
| Involvement or engagement | advocacy OR collaborat* OR co*production OR consult* OR empower* OR engage* evaluat* OR involv* OR particip* OR partner* OR PPI OR organi*ation* OR representation* |
| Big data | database OR "big data" OR "data science" OR "data mining" OR "datasets" OR "data analytics" OR "data sets" |
| Public involvement | "patient participation" OR "consumer participation" OR "client participation" OR "community participation" |
| Data governance (only Health Research Premium Collection, Scopus and Web of Science) | "data governance" |

## Big data

Data which is challenging to manage through traditional analytic tools and meets the 5V characteristics: volume, velocity, variety, veracity and value.[1]

The volume suggests that there may be a high quantity of data available potentially on millions of patients. The variety means heterogeneity of data collected as it can come in various formats (eg, images, text). The velocity means that it can be collected swiftly from various sources. Veracity relates to the accuracy and identification of any biases. The value refers to the ability of results from research based on big data to guide decisions. Big data sources can be internal (eg, patients record, healthcare professional notes, generated through apps or social media) and external (eg, private companies or governmental institutions).

To map a range of studies, we will keep the selection criteria purposefully broad. Papers can discuss single research project or data sharing initiative. All study designs will be included. Papers can be (but not limited to) original research, an evaluation, a review, an expert opinion or a commentary that explores any public involvement and engagement in big data research.

We will exclude a paper if it:
▶ Does not discuss public involvement or engagement.
▶ Does not discuss a patient-related (or health-related) application.
▶ The full text is not available in English.

## Study selection

Prior to the screening stage, we will organise a meeting for everyone involved in the study selection process during which we will jointly scan a sample of 100 papers. We will record and discuss our disagreements. Then separately, we will scan all papers' eligibility, based on the title and then the abstract identified in the database searches. At each stage, two reviewers will be involved. The first reviewer will scan all papers and the second will check a random sample (20% of all papers). Reasons for exclusion will be recorded. If there are any disagreements, we will include a third reviewer. Then the full text will be screened by two reviewers, checking if the paper meets the inclusion and exclusion criteria. We will meet after each screening stage (title, abstract and full paper) to discuss our experiences.

## Data extraction

We will use an extraction form which will cover the following information:

► Paper aim.
► Design.
► Country.
► Demographics of participants (also record if there are a seldom-heard group).
► Context.
► Process of involvement or engagement.
► Funding.
► Legal or ethical issues.
► References to guidance and policies.
► Challenges and facilitators of public involvement and engagement.

We see the extraction stage as an iterative process. After extracting initial papers, we will discuss if the extraction form is applicable in our review during team meetings. Where necessary, we will revise it. Each paper will be extracted by one reviewer and the second will validate data extraction.

## Reporting the results

We will provide a descriptive and narrative analysis of the data. These will be used to develop the system model. Then, we will discuss the implication of the findings for researchers and policy.

## Stage 2: systematic review
### Criteria for inclusion

Out of papers identified in the scoping review, we will extract qualitative and quantitative studies that discuss the delivery or effectiveness of involvement and engagement strategies.

## Data extraction and synthesis

We will follow Thomas and Harden[36] stages of qualitative synthesis. We plan to extract all findings sections from included papers and upload them to NVivo for analysis. Coding will be done inductively to develop descriptive themes to further our review aims and develop the system model. Thus, we want to ensure that no prior framework will influence us in identifying the relevant evidence. The relevant results from the quantitative studies will be extracted and placed under qualitative themes, as we do not expect that meta-analysis will be possible. At the last stage of the synthesis, we go beyond the descriptive themes and analyse them in the context of the aims of our review. The results will be provided in a thematic, narrative way and used to develop the system model.

## Studies and findings appraisal

Using Mixed Methods Appraisal Tool (MMAT)[37] we will systematically appraise all studies included in the systematic review. However, no paper will be excluded if it scored low. The overall confidence in each individual qualitative findings will be assessed through the Confidence in the Evidence from Reviews of Qualitative Research (GRADE-CERQual).[38] We will not assess the overall confidence in quantitative studies as these will be placed under the qualitative themes. This will allow researchers to make judgements about the quality of available evidence.

## Patient and public involvement

Stakeholders (including patients and health professionals) can be involved in systematic reviews.[39 40] They can enhance the quality of the review by advising on the review questions and its scope. This ensures transparency and accountability, especially if the review aims to shape practice and improves relevance to those who this review seeks to influence (eg, practitioners and public). Similarly, for scoping reviews. Arksey and O'Malley[27] recommend, and Levac *et al*[29] argue that consultation is a part of the review process. We have involved two public advisors who assisted in designing this protocol and will be coauthors on all publications. They have experience of conducting systematic reviews, represent seldom-heard communities and SA is a Big Data Ambassador for Care and Health Informatics theme within ARC NWC. They will be involved in the whole review process, with a particular emphasis on interpreting the findings and developing recommendations for both research and practice. We will report on public involvement using the Guidance for Reporting Involvement of Patients and the Public (GRIPP2) checklist.[41]

## Limitations

The main limitation of our review is the exclusion of non-English papers. There is a possibility that some papers relevant to our review aims will be excluded and this will impact our findings. Second, as already mentioned the lack of clear definitions of public involvement, engagement and big data make any search strategy challenging, and potentially some relevant papers might not be included. However, we will undertake all reasonable steps to balance this limitation by involving experts and checking references in included papers.

## Ethics and dissemination

We have published this protocol and engaged with public advisors to ensure transparency and rigour of our review

process. As we are using already published data, there is no need to apply for ethical approval to conduct our study. We will present our findings at relevant conferences and publish in a peer-reviewed journal.

## DISCUSSION

This review will synthesise the current literature on public involvement and engagement in big data research. Our work is timely as it is expected that big data research in healthcare will continue to grow rapidly. There will be increasing interest in developing large health data sets by researchers, funders and governmental bodies. Previous research shows the need for synthesising the current evidence. Mouton et al[42] discussed issues around patient trust and big data, and how they viewed healthcare practitioners and professionals' involvement in funding or controlling big data research. They believed that patients were not interested or did not understand big data—and therefore, should not be involved in its governance. Their comments included remarks that patient groups are not important and the belief that patients' involvement in governance would be pointless. On the other hand, Aitken et al[8] explored the similar issues with members of the public who presented opposite views on lay involvement in data governance. Participants believed that members of the public could promote accountability of big data research. Public involvement has the potential to shift perspectives and bridge the gap between researchers and the public, and help the development of big data research that has wider spread public support and buy-in.

**Acknowledgements** The development of the search strategy was supported by Zoe Gibbs-Monaghan Liaison Librarian—Health and Life Sciences at the University of Liverpool. Ruaraidh Hill, Lecturer in evidence synthesis at the University of Liverpool provided beneficial insights into review's methodology.

**Contributors** PT developed the study design, drafted the protocol and conducted initial searches with the assistance of the librarian. EJ, NT, SA, LF contributed to drafting and editing. All authors have read and approved the final manuscript.

**Funding** PT is a PhD student supported by the National Institute for Health Research Applied Research Collaboration North West Coast (NIHR ARC NWC) Award number: 103. The views expressed in this publication are those of the author(s) and not necessarily those of the National Institute for Health Research or the Department of Health and Social Care.

**Competing interests** None declared.

**Patient and public involvement** Patients and/or the public were involved in the design, or conduct, or reporting, or dissemination plans of this research. Refer to the Methods section for further details.

**Patient consent for publication** Not required.

**Provenance and peer review** Not commissioned; externally peer reviewed.

**ORCID iDs**
Piotr Teodorowski http://orcid.org/0000-0003-2172-8298
Lucy Frith http://orcid.org/0000-0002-8506-0699

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
