## [Reviewer comments · BMJ Open]

ARTICLE DETAILS

TITLE (PROVISIONAL)	Public involvement and engagement in big data research: protocol for a scoping review and a systematic review of delivery and effectiveness of strategies for involvement and engagement.
AUTHORS	Teodorowski, Piotr; Jones, Elisa; Tahir, Naheed; Ahmed, Saiqa; Frith, Lucy

VERSION 1 – REVIEW

REVIEWER	Leightley, Daniel King's College London
REVIEW RETURNED	12-Apr-2021

GENERAL COMMENTS	The authors present a protocol for a scoping review of public involvement and engagement in big data research projects. The review is timely considering the changes in data landscape as the BREXIT data governance transition takes place, and our growing reliance on big data. General comments: 1. It needs to be clearer on the two stages of the study; and how the 'scoping review' and 'systematic review' are substantially different (and complement each other). At present, the protocol reads as if the authors have chosen the current approach to be lax in the 'systematic' element, and to enable greater flexibility in the synthesis. This is acceptable, but it would be easier to state this.2. Depending on the above response, it is still possible to register the 'systematic review' part on Prospero.3. The authors state that the protocol will be conducted to ensure rigour and transparency, yet state that data sharing is not applicable. It would be helpful to be open to sharing coder sheets. Specific comments: Page 2: Line 56/57: While regional projects are useful, are the authors not able to cite any national projects that represent big data on a country-based scale? Page 3: Line 10: The authors state 'despite these ethical issues', but it isn't clear what these ethical issues are. Several factual statements are made prior to this statement; however, they don't raise any direct ethical issues. Please be explicit. Page 4: Line 8: Reference formatting error. Page 4: Line58: What is the rationale for selecting 2010 as the start date for the search? Can the authors support this decision?
---

	Page 5: Public search terms: In addition to the terms provided, another common term used is steering group. Page 6: Line 41: Please state if you will provide an inter rater reliability in the published article. Further, will the second rater screen abstract, full text or validate data extraction? Page 6: Big data search terms: What about terms relating to database, which is a traditional term used to represent big data. Page 7: PPI: It is excellent to see the authors gain support from a PPI group. However, it could be useful to provide a little context as to why these two individuals have been selected to advise on this project. Do they have specific domain expertise?
--	--

REVIEWER	Kolstoe, Simon University of Portsmouth, School of Healthcare Professions
REVIEW RETURNED	13-May-2021

GENERAL COMMENTS	I thought this presented an excellent justification and summary of the methods you will use in a clearly interesting and timely topic. The only suggestion I would have is to maybe look at (and if applicable reference) the Health Research Authorities "Make It Public" campaign/activities and also the NIHR guidance on Public and Patient Involvement (https://www.nihr.ac.uk/documents/ppi-patient-and-public-involvement-resources-for-applicants-to-nihr-research-programmes/23437). Good luck with the reviews!
---

VERSION 1 – AUTHOR RESPONSE

Reviewer: 1

Dr. Daniel Leightley, King's College London Comments to the Author:

The authors present a protocol for a scoping review of public involvement and engagement in big data research projects. The review is timely considering the changes in data landscape as the BREXIT data governance transition takes place, and our growing reliance on big data.

Thank you for this comment and for recognising the importance of conducting this review.

General comments:

1. It needs to be clearer on the two stages of the study; and how the 'scoping review' and 'systematic review' are substantially different (and complement each other). At present, the protocol reads as if the authors have chosen the current approach to be lax in the 'systematic' element, and to enable greater flexibility in the synthesis. This is acceptable, but it would be easier to state this.

Page 4, line 26- We provided details on the benefits of the scoping reviews around identifying gaps in the literature and the systematic review on the ability to rigorously synthesise literature around a specific issue (which is delivery and effectiveness of strategies in our case) to provide recommendations for practitioners. Now, we also include additional reference (Munn, et al. 2018) to support our arguments behind the choice of scoping and systematic reviews in our review process. We also add that these reviews complement each other and give us flexibility in understanding this unexplored phenomenon.

2. Depending on the above response, it is still possible to register the 'systematic review' part on Prospero.

Based on the above response, we will not register the systematic review part on Prospero. This also reflects the inclusion of PRISMA-ScR checklist for scoping reviews instead of PRISMA-P for systematic reviews.

3. The authors state that the protocol will be conducted to ensure rigour and transparency, yet state that data sharing is not applicable. It would be helpful to be open to sharing coder sheets.

Page 9, line 16- Data statement refers to the protocol. We clarify that as there is no datasets created or analysed in this protocol, data sharing is not applicable. We are also open to share coder sheets when review findings will be available.

Specific comments:

Page 2: Line 56/57: While regional projects are useful, are the authors not able to cite any national projects that represent big data on a country-based scale?

Page 2 line 49- We remove one reference of the regional project and add the national project OpenSAFELY instead as an example. We hope this will provide a reader with helpful examples of how much this kind of work has developed in the UK over the last couple of years.

Page 3: Line 10: The authors state 'despite these ethical issues', but it isn't clear what these ethical issues are. Several factual statements are made prior to this statement; however, they don't raise any direct ethical issues. Please be explicit.

Page 3 line 8 The previous sentences provided background to ethical challenges. To link them better with the sentence, the reviewer mentioned, we also summarised them.

Page 4: Line 8: Reference formatting error.

The reference has been corrected.

Page 4: Line58: What is the rationale for selecting 2010 as the start date for the search? Can the authors support this decision?

Page 4, line 50- The rationale is that big data is a newly developing field. The argument has been mentioned in the prior sentence. We rephrase it, making this connection clearer.

Page 5: Public search terms: In addition to the terms provided, another common term used is steering group.

Page 5, line 10- We add "steering group*" as additional search term.

Page 6: Line 41: Please state if you will provide an inter rater reliability in the published article. Further, will the second rater screen abstract, full text or validate data extraction?

Page 6 line 37- We clarify that before the screening stage, we will organise a meeting bringing everyone involved in the screening process and jointly review a sample of 100 papers. Then, we will record and discuss our disagreements. Also, further information has been added to clarify that two reviewers will be involved (including later in the data extraction section).

Page 6: Big data search terms: What about terms relating to database, which is a traditional term used to represent big data.

Page 5, line 10- We added the database as an additional search term. Other similar terms include "datasets" or "data sets"

Page 7: PPI: It is excellent to see the authors gain support from a PPI group. However, it could be useful to provide a little context as to why these two individuals have been selected to advise on this project. Do they have specific domain expertise?

Page 7, line 47- We added the following sentence clarifying public advisors experiences and background: "They have experience of conducting systematic reviews, represent seldom-heard communities and SA is a Big Data Ambassador for Care and Health Informatics theme within ARC NWC."

Reviewer: 2

Dr. Simon Kolstoe, University of Portsmouth Comments to the Author:

I thought this presented an excellent justification and summary of the methods you will use in a clearly interesting and timely topic.

Thank you so much for this positive comment.

The only suggestion I would have is to maybe look at (and if applicable reference) the Health Research Authorities "Make It Public" campaign/activities and also the NIHR guidance on Public and Patient Involvement (<https://www.nihr.ac.uk/documents/ppi-patient-and-public-involvement-resources-for-applicants-to-nihr-research-programmes/23437>).

This is a comprehensive list of resources. We decided to reference the GRIPP2 checklist (Page 8, line 2), which we will utilise throughout the review process to improve the way we will report on how public involvement shaped the quality of the review.

VERSION 2 – REVIEW

REVIEWER	Leightley, Daniel King's College London
REVIEW RETURNED	07-Jul-2021
GENERAL COMMENTS	The authors have addressed my concerns.